# Oxytocin Administration in Low-Risk Women, a Retrospective Analysis of Birth and Neonatal Outcomes

**DOI:** 10.3390/ijerph18084375

**Published:** 2021-04-20

**Authors:** Xavier Espada-Trespalacios, Felipe Ojeda, Mercedes Perez-Botella, Raimon Milà Villarroel, Montserrat Bach Martinez, Helena Figuls Soler, Israel Anquela Sanz, Pablo Rodríguez Coll, Ramon Escuriet

**Affiliations:** 1Department of Obstetrics and Gynecology, Hospital General de Granollers, Avinguda Francesc Ribas s/n, 08402 Granollers, Spain; fojeda@fphag.org (F.O.); mbach@fphag.org (M.B.M.); hfiguls@fphag.org (H.F.S.); 2Department of Experimental and Health Sciences, Universitat Pompeu Fabra (UPF), Doctor Aiguader 88, 08003 Barcelona, Spain; 3Research Group in Global Health, Gender and Society (GHenderS), Universitat Ramon Llull, Carrer Padilla 326, 08025 Barcelona, Spain; rescuriet@gencat.cat; 4Research in Childbirth and Health Unit (ReaRH), University of Central Lancashire, Preston PR1 2HE, UK; MPerez-Botella1@uclan.ac.uk; 5Department of Neonatology, Hospital General de Granollers, Avinguda Francesc Ribas s/n, 08402 Granollers, Spain; 6School of Health Sciences Blanquerna, Universitat Ramon Llull, Carrer Padilla 326, 08025 Barcelona, Spain; raimonmv@blanquerna.url.edu (R.M.V.); ianquela@fphag.org (I.A.S.); 7Obstetric Care Area, Hospital Germans Trias i Pujol, Carretera de Canyet s/n, 08916 Badalona, Spain; pablo.rodriguez.coll@gmail.com; 8Catalan Health Service, Government of Catalonia, Travessera de les Corts 131, 08028 Barcelona, Spain

**Keywords:** term birth, obstetric labor, oxytocin, low-risk pregnancy, birth outcome, neonatal outcome

## Abstract

Background: In recent years, higher than the recommended rate of oxytocin use has been observed among low-risk women. This study examines the relationship between oxytocin administration and birth outcomes in women and neonates. Methods: A retrospective analysis of birth and neonatal outcomes for women who received oxytocin versus those who did not. The sample included 322 women with a low-risk pregnancy. Results: Oxytocin administration was associated with cesarean section (aOR 4.81, 95% CI: 1.80–12.81), instrumental birth (aOR 3.34, 95% CI: 1.45–7.67), episiotomy (aOR 3.79, 95% CI: 2.20–6.52) and length of the second stage (aOR 00:18, 95% CI: 00:04–00:31). In neonatal outcomes, oxytocin in labor was associated with umbilical artery pH ≤ 7.20 (OR 3.29, 95% CI: 1.33–8.14). Admission to neonatal intensive care unit (OR 0.56, 95% CI: 0.22–1.42), neonatal resuscitation (OR 1.04, 95% CI: 0.22–1.42), and Apgar score <7 (OR 0.48, 95% CI: 0.17–1.33) were not associated with oxytocin administration during labor. Conclusions: Oxytocin administration during labor for low-risk women may lead to worse birth outcomes with an increased risk of instrumental birth and cesarean, episiotomy and the use of epidural analgesia for pain relief. Neonatal results may be also worse with an increased proportion of neonates displaying an umbilical arterial pH ≤ 7.20.

## 1. Introduction

Oxytocin is an endogenous hormone, produced in the hypothalamus and secreted by a feedback mechanism by the pituitary gland [1]. Its role during labor and birth is to generate rhythmic contractions of the uterine smooth muscle to ensure progressive dilatation and eventually the birth [2].

Synthetic oxytocin can be administered to commence or increase uterine activity during labor. Administering exogenous oxytocin while the endogenous form is being released can overwhelm the feedback mechanism leading to an ineffective regulation of uterine activity with potentially disastrous consequences for the mother and fetus [3].

Oxytocin is considered a high-risk drug by the Institute for Safe Medication Practices. High-risk drugs are those that, when used inappropriately, have the highest potential to cause severe harm to patients [4].

The Normal Birth Care Strategy published in 2008 by the Spanish Ministry of Health establishes standards for intrapartum care in Spain for women with a low-risk pregnancy and labor [5]. It recommends that no more than 10% of women in this group should undergo induction of labor (IOL) or augmentation of labor (AOL).

There is a paucity of data on the use of oxytocin among low-risk women and the available evidence shows wide variations in its use. In Spain, a study to evaluate the implementation of the Normal Birth Care Strategy shows a rate of 19.4% IOL and 53.3% AOL, amply surpassing its recommendations [6]. A Swedish study observed a 33% oxytocin use [7], while a French study showed a rate above 60% [8] and in the United States, Iobst et al. reported that 44% of low-risk nulliparous women had oxytocin administered during labor [9].

One of the explanations as to why rates of oxytocin use vary so widely may lie in the fact that specifying the reason for administration of oxytocin in the woman’s records is not globally or homogeneously recorded (i.e., using the same codes or diagnostic criteria for example to diagnose slow labor). The reason for oxytocin administration should be identified in each case, and close monitoring should be implemented throughout the process, to observe for uterine response in each woman. Based on the reason for administration, the lowest possible dose must be used, and it should not be increased earlier than every 30 min. This will allow us to assess whether the current dose is sufficient or whether further oxytocin is needed [10].

International scientific bodies have established the indications for oxytocin administration during labor [11,12,13]. However, several studies suggest excessive and unnecessary use of oxytocin intrapartum: rather than treating labor dystocia, oxytocin is being used to prevent it with the potential increase in unnecessary complications [14,15].

Oxytocin administration during labor is associated with negative obstetric outcomes, such as an increase in the rate of the cesarean section following IOL [16], an increase in instrumental birth and episiotomy following AOL [14,15,17], and an increased risk of postpartum hemorrhage (PPH) [18]. Oxytocin administration is also associated with poorer neonatal outcomes such as an increased risk of admission to the neonatal intensive care unit (NICU) [19,20], neonatal acidemia [21,22], lower Apgar scores, and greater need for neonatal resuscitation [7,15,23].

Furthermore, a Swedish study by Ekelin et al. found that the majority of midwives they interviewed thought that the use of oxytocin to accelerate labor was being used excessively and unnecessarily in most cases [24].

This backdrop seems to indicate that the use of oxytocin may be guided by a desire to prevent specific undesirable maternal and neonatal outcomes, without paying due attention to the added, unintended negative consequences that using oxytocin can bring and which are well described within the scientific literature [25]. It is therefore imperative that clear guidelines are developed for the appropriate use of oxytocin for IOL and AOL.

The idea for our study stems from the higher than recommended rate of oxytocin use among low-risk women within our setting.

Before we can put in place measures to ensure more judicious use of oxytocin in labor, we must first gain a better understanding of what are the factors and circumstances of its use, including the reasons and situations in which it is being employed. We should also compare our results with similar studies, as well as assess whether we comply with international recommendations. Finally, we must evaluate the impact that oxytocin use can have on women and neonates.

With this in mind, the objective of our study is to analyze the use of oxytocin in low-risk women in our setting and its impact on maternal and neonatal outcomes. The results of this study will help inform strategies and measures to provide a more physiological approach to labor care.

## 2. Materials and Methods

This is a retrospective observational study with prospectively collected data.

The study population is comprised of women who gave birth at the General Hospital of Granollers between 1 October 2016 and 1 April 2017. The birth rate at the hospital is around 1500 births per year.

Consecutive enrolment was carried out for those participants meeting inclusion criteria until the end of the recruitment period. Data were entered in a pre-designed data collection sheet and they were then inputted into an electronic database.

Inclusion criteria: pregnant women between 18 and 40 years old, with a single, cephalic presentation, low-risk pregnancy at the time of admission in labor according to prenatal care protocol in Catalonia [26], which starts between 37 + 0 and 41 + 6 weeks of gestation.

Exclusion criteria: pregnant women 17 years or younger or 41 years or older; multiple pregnancies; women who were admitted for an elective cesarean section; any fetal presentation other than cephalic; women not classified as low-risk at the time of admission in delivery suite according to prenatal care protocol in Catalonia and a start of labor before 37 + 0 or after 41 + 6.

Prenatal care protocol in Catalonia defines a woman with a low-risk pregnancy as a pregnant woman who has a physical characteristic, a physiological history or a pathology that does not require resources or specialized care. The criteria that define the gestational risk classification are well specified in the protocol and the risk classification is carried out continuously by the midwife during the antenatal follow-ups (Figure 1).

In our country, delivery care for women with low-risk pregnancy is mostly performed by midwives. Women in Catalonia cannot choose an elective IOL on their own, as IOL is indicated by an obstetrician under a woman’s informed consent.

Including in the study low-risk women who had an IOL may seem contradictory, as induction itself renders the pregnancy and labor high-risk. However, healthy women who have an elective IOL for no medical reason [27] represent a considerable proportion of laboring women, and it is thought to be on the increase [28]. It is estimated that this group of women could make up around 20% of all IOL [29,30]. A separate analysis for this group of women was carried out to identify the added risks these women were subjected to.

Participants’ socio-demographic data were collected, as well as data about maternal, birth and neonatal outcomes.

At the Hospital General de Granollers, an obstetrician must be consulted before initiating oxytocin infusion. The prescribed dose of oxytocin infusion is 5 IU in 500 mL physiological saline with an initial dose of 2 mU/min, the dose is increased by 2 mU/min in AOL and in IOL is doubled until it reaches 8 mU/min. Then it is increased by 2 mU/min until five contractions in 10 min are reached or until the maximum dose of 20 mU/min.

The main variable under study is the administration of oxytocin during the first and second stages of labor, as the administration of 5 IU of oxytocin during the third stage of labor for placental delivery is common practice in the study setting.

Birth outcome measures included instrumental birth, cesarean section, episiotomy and PPH (blood loss >500 mL).

Neonatal outcome measures comprised an Apgar of <7 at 1 and 5 min of life, the need for neonatal resuscitation at birth, admission to NICU and arterial umbilical pH ≤ 7.20.

We conducted a comparative analysis of birth and neonatal results between women who had oxytocin administered in labor and those who did not. Women who received oxytocin did so intending to induce or accelerate labor. We also assessed whether the neonatal and birth results were associated with the reason for oxytocin administration. All women in this study received intravenous oxytocin when augmented or induced.

Dependent variables were the mode of delivery, presented as cesarean section, instrumental vaginal birth and spontaneous birth, postpartum hemorrhage >500 mL, episiotomy, length of first and second stage labor, neonatal resuscitation, umbilical artery Ph ≤ 7.20, transfer to the neonatal intensive care unit and Apgar score < 7 at one and five minutes. Independent variables were oxytocin administration and indication to oxytocin administration (IOL versus AOL)

For quantitative variables following a normal distribution, we used the mean and standard deviation. To analyze non-parametric variables, we used the median (1–3 quartile range). Quantitative variables were compared using Student’s *t*-test. Non-parametric tests were used for the variables of first and second stage lengths as they did not follow a normal distribution; this analysis was only conducted for vaginal births as no data was available for cesarean births.

The Odds Ratio (OR) was calculated with a CI of 95%. The variables selected to conduct the Odds Ratio analysis were identified based on their possible impact on birth and neonatal outcomes as per available scientific evidence, for example, the effect of an epidural on cesarean section [31]. Adjusting variables were parity, peridural analgesia, birth weight, gestational age, vaginal birth and oxytocin indication. *p* < 0.05 was considered statistically significant. The first stage length was estimated from the onset of the active phase or admission to the labor ward if they had already reached this phase. In the cesarean section, if the first and/or second stage of labor was not completed, these lengths were not analyzed. Finally, a univariate logistic regression was conducted to establish associations between the variables; to adjust for confounding factors, multivariate logistic regression was carried out.

When adjusting neonatal results per type of birth, cesarean sections were excluded from the analysis to avoid bias: for those cesareans performed for fetal compromise, it would not have been possible to determine if administration of oxytocin was the cause of such compromise.

We did not conduct a multivariate analysis for those variables with few cases (Apgar score <7 at 5 min, 3 cases; 3rd and 4th-degree perineal damage, 4 cases; umbilical artery pH < 7.10, 6 cases; cesarean section in multiparous women without oxytocin, 0 cases; PPH in primiparous women without oxytocin, 0 cases).

Ethical Statement: The project was approved by the Ethics Committee (Clinical Research Ethics Committee of Hospital de Granollers) on 6 June 2016 (approval number: 20162012) and was also approved by the board of directors of the hospital as it is customary in our setting. The ethics committee of the hospital required consent from the women and written consent was obtained. Further information and documentation are available on request.

## 3. Results

Of a total of 755 births during the study period, 329 women met the inclusion criteria, 6 women refused to participate in the study, and 1 woman was excluded from study owing to incomplete data. The final study population was 322 women (Figure 2).

Out of the 322 women, a total of 232 (72%) received oxytocin during labor. The start of labor was spontaneous in 78.9% of the women and 21.1% had an IOL. A total of 64.6% of the women who started labor spontaneously received oxytocin to accelerate labor while only 35.4% did not receive any oxytocin.

The mean age of the participants in the study was 30 years old. The mean age for those who were administered oxytocin was 31 years old while it was 30 for those who did not receive oxytocin.

We analyzed the relationship between oxytocin administration and several maternal and neonatal characteristics (Table 1). 

Oxytocin was administered in a greater proportion to women with the following characteristics: Spanish nationality (65.8%), secondary studies (36.2%), 40 weeks gestational age at the time of birth (30.6%), nulliparous women (60.3%) and those with neonates weighing between 3000 g and 3500 g (45.7%)

Women who were administered oxytocin had longer first and second stages of labor (46 min; *p* = 0.01, and 18 min; *p* < 0.01 respectively).

The most prevalent type of birth in this group of women was spontaneous vaginal (66.7%), followed by an instrumental (22%) and finally a cesarean section (8.2%).

We found the greatest proportion of women who had been administered oxytocin among those women who had a cesarean section (90.5%), followed by those who had an instrumental birth (87.9%). These differences were statistically significant (*p* < 0.01).

Overall, women who had oxytocin administered during labor sustained more interventions such as epidural analgesia (EA) (97.8% of women; *p* < 0.01), artificial rupture of membranes (ARM) (57.3% of women, although this did not reach statistical significance; *p* = 4.08) and episiotomy (58.7% of women; *p* < 0.01).

With regard to PPH, this occurred in a greater proportion among those who received oxytocin (5.6% versus 1.1% among those who did not receive it). However, these values reached no statistical significance (*p* = 0.08).

On the other hand, there was a greater percentage of neonates whose mothers received oxytocin during labor who scored <7 in the Apgar test at 1 min of life (56.3%; *p* = 0.15). A score <7 at 5 min was more prevalent among those neonates whose mothers did not receive oxytocin (66.7%; *p* = 0.13), although only 3 cases were reported.

Mothers whose neonates required resuscitation were more likely to have been administered oxytocin (72.7%; *p* = 0.96). Neonates admitted to NICU were in greater proportion born to mothers who had been administered oxytocin too (60%; *p* = 0.22). The most significant difference was between the administration of oxytocin and a pH ≤ 7.20. This was more prevalent among those neonates whose mothers had received oxytocin in labor (87.8%; *p* = 0.01).

We conducted a logistic regression (Table 2) to understand the impact that oxytocin administration had on birth outcomes. 

This showed a non-significant increase in cesarean sections. However, when we adjusted for confounders, it became significant (aOR 4.81; CI 95% 1.80–12.82; *p* < 0.01).

When we adjusted the results of other birth variables for confounding, we did not observe differences between crude and adjusted OR. There was an increased probability of labor resulting in an instrumental birth (aOR 3.34; CI 95% 1.45–7.67; *p* < 0.01). Likewise, episiotomy was more readily performed in the presence of oxytocin (aOR 3.79; CI 95% 2.20–6.52; *p* < 0.01). PPH had a positive association among those women who received oxytocin; however, this did not reach statistical significance (aOR 1.55; CI 95% 0.49–4.91; *p* = 0.46).

Equally, the length of the first stage of labor was increased in the presence of oxytocin (by 46 min), but this was not statistically significant (*p* = 0.05). In contrast, the length of the second stage of labor increased by 18 min and this did reach statistical significance (*p* < 0.01).

Following adjustment for parity (Table 3), we observed that nulliparous women who received oxytocin had a positive association with cesarean section (OR 1.08; CI 95% 0.88–1.32; *p* = 0.54) and instrumental birth (OR 1.14; CI 95% 1.00–1.30; *p* = 0.09) although neither reached statistical significance. We could not assess the risk of cesarean section among multiparous women due to a lack of cases in the group who did not receive oxytocin.

The use of EA in the presence of oxytocin was influenced by parity: multiparous women used it more frequently than nulliparous women (OR 9.63; CI 95% 3.24–28.61; *p* < 0.01 versus OR 4.29; CI 95% 1.24–14.83; *p* < 0.01).

With regard to the incidence of episiotomy in the presence of oxytocin, primiparous women were at higher risk than multiparous women (OR 1.75; CI 95% 1.06–2.86; *p* < 0.01 versus OR 1.66; CI 95% 1.18–2.33; *p* < 0.01).

We could not assess the risk of PPH among primiparous women due to a lack of cases in the group who not received oxytocin and for multiparous women, we saw an increased risk, albeit not a statistically significant one (OR 1.39; CI 95% 0.95–2.03; *p* = 0.25).

The length of the first and second stages of labor did not show statistical significance between nulliparous and multiparous women who received oxytocin in labor.

We also adjusted the OR by reason for administration (IOL and AOL) (Table 3). When oxytocin was administered for IOL, the risk of cesarean section increased significantly (OR 2.20; CI 95% 1.64–2.96; *p* < 0.01). The risk of instrumental birth increased when oxytocin was administered for IOL or AOL (OR 1.69; CI 95% 1.17–2.44; *p* = 0.02 versus OR 1.39; CI 95% 1.18–1.65; *p* < 0.01).

Administration of oxytocin for either IOL or AOL had a positive association with the use of EA (IOL: OR 4.54; CI 95% 1.36–15.11; *p* < 0.01) (AOL: OR 32.26; CI 95% 1.67–135.68; *p* < 0.01). Furthermore, the positive association was also shown with episiotomy when oxytocin was administered for IOL or AOL (OR 2.82; CI 95% 1.55–5.14; *p* < 0.01 versus OR 2.12; CI 95% 1.53–2.95; *p* < 0.01). Both IOL and AOL increased the risk of PPH although the values only showed statistically significant results in IOL (OR 2.01; CI 95% 1.34–3.01; *p* = 0.04).

Women who had oxytocin for AOL had the first stage of labor 56 min longer than those women who did not receive oxytocin (*p* = 0.03) and the second stage of labor was 23 min longer (*p* <0.01).

We also adjusted for use of EA and the results indicate that the use of this in women who received oxytocin during labor did not increase the risk of cesarean (*p* = 0.05) or PPH (*p* = 0.08). However, it did show positive association in instrumental births (OR 3.34; CI95% 1.45–7.67; *p* < 0.01) and episiotomy (OR 3.79; CI95% 2.20–6.52; *p* < 0.01).

EA also increases the length of the first stage of labor, but not in a statistically significant manner (*p* = 0.56). On the other hand, EA increases the length of the second stage of labor by 44 min (*p* < 0.01), but only among those women who did not receive oxytocin.

The logistic regression for neonatal results showed that when oxytocin was administered, the probability of obtaining a pH ≤ 7.20 from the umbilical artery was greater than when oxytocin was not used (OR 3.29; CI 95% 1.33–8.14; *p* < 0.01). The probability to obtain a score of <7 in the Apgar test at one minute of life (OR 0.48; CI 95% 0.17–1.33; *p* = 0.15) did not increase significantly when oxytocin was administered. This was the same for neonatal resuscitation (OR 1.04; CI 95% 0.22–1.42; *p* = 0.96) and the need for admission to NICU (OR 0.56; CI 95% 0.22–1.42; *p* = 0.22).

We adjusted OR by type of vaginal birth with oxytocin administration during labor as a predictor (Table 4). 

We observed that instrumental birth had a negative association with Apgar test <7 at one minute of life (aOR 0.05; CI 95% 0.01–0.43; *p* < 0.01), and also spontaneous birth showed a negative association with neonate admission to NICU (aOR 0.24; CI 95% 0.06–0.97; *p* = 0.05). However, low umbilical artery Ph (≤7.20) and the need for neonatal resuscitation did not reach statistical significance.

Finally, we calculated the OR for the reason of oxytocin administration (Table 4). IOL and AOL had a positive association for umbilical artery Ph ≤7.20 (aOR 3.71; CI 95% 1.32–10.46; *p* = 0.01 and aOR 3.12; CI 95% 1.23–7.95; *p* = 0.02 respectively). IOL showed a positive association with neonatal resuscitation and NICU admission, but without statistical significance.

## 4. Discussion

Our study population comprised low-risk women, yet the rates of oxytocin use are higher than those reported by other studies [8,14]. Furthermore, they are higher than those recommended by the Normal Birth Care Strategy [5] which recommends that AOL for women who start labor spontaneously should stand between 5–10%. In stark contrast, our results show 50.9% of women were augmented with oxytocin during labor. With regard to IOL, the strategy considers that less than 10% of labors should be induced, while our rates show an incidence of 21.1%, similar to other studies which have quantified elective IOL for low-risk women [29,30].

Women in our study who received oxytocin in labor were more likely to require EA than those who did not receive it. These results are similar to those reported by other studies which showed a correlation between the use of oxytocin in labor (either for IOL or AOL) and the use of EA [14,19,32,33,34].

However, the rates reported by those studies were much lower than the ones we obtained in our study where 85% of all women used EA, and this was even higher among those who received oxytocin in labor: 97.8%. Current recommendations suggest EA should stand between 30–80% of women [5].

The available literature offers conflicting evidence on the association between EA and poor obstetric outcomes [34,35]. When parity was included in the regression analysis for the group of women who received oxytocin in our study, the adjusted OR showed that the use of EA was higher among multiparous women. Both nulliparous and multiparous women had a high EA use.

Despite this strong link between the administration of oxytocin and EA use in our study, we are unable to discern whether high EA use is the consequence of oxytocin administration in labor (because it is a more painful process) or because women who are administered oxytocin are more fearful of painful labor and request EA more readily.

ARM was conducted more frequently in the group of women who received oxytocin, but this did not reach statistical significance. The overall rate of ARM in our study was 55.9%, whereas ARM among women who received oxytocin was 1.4% higher. This may point to a routine overuse of an unnecessary intervention for low-risk women, which contravenes national and international recommendations [5,11,36,37,38]. Furthermore, routine use of this intervention has proven to be ineffective in preventing labor dystocia [12].

The duration of the first and second stages of labor in women receiving oxytocin was longer. This represents a puzzling paradox: attempting to reduce the length of labor through the administration of oxytocin may lead to longer labors. According to a systematic review comparing low doses versus high doses of oxytocin, a shortening of the first stage of labor was only achieved when oxytocin was administered in high doses [39]. Based on this premise, we analyzed the type of oxytocin doses administered in our center and we saw that they were low according to Kenyon et al.’s classification (low-dose regimens: defined as starting dose of 4 mU per minute and an increment of less than 4 mU per minute with an increased interval between 15 and 40 min).

While some studies have shown that IOL is associated with longer first and second stages of labor [40], our results are contrasting and show a reduction of these times, however, these were not statistically significant. On the other hand, we observed an increase in the length of labor in the AOL group. When we analyzed the length of the first and second stages of labor and adjusted the OR we saw that EA is not a confounding factor, while parity is.

Regarding cesarean section, we observed that oxytocin was not a risk factor, but when we adjusted the OR by reason for its administration, we saw that when used to induce labor, it carried a significantly higher risk for cesarean. The associated risk between IOL and cesarean section has already been reported in other studies [16,19]. Conversely, these findings disagree with the latest Cochrane reviews and others [41,42] who compare IOL with expectant management of labor. Studies whose results are consistent with ours compare IOL with spontaneous onset of labor. Therefore, the discrepancy between studies may be due to the methodology used.

The association has also been found even when controlling for pregnancy risk [20] and other different interventions associated with oxytocin administration [9]. 

When EA was administered to women receiving oxytocin, the risk of cesarean was not increased. These findings are in line with a recent systematic review [43] which found no relationship between EA and cesarean section.

Some studies have found a relationship between instrumental births and oxytocin administration when a misdiagnosis of labor dystocia had been made [14,15]. The level of risk for instrumental birth when oxytocin is used to augment the labor in our study is similar to those found by a Norwegian study [17] and with similar rates of AOL. However, we obtained higher risk levels than those reported by other studies [14,15]. This may be explained by the higher incidence of AOL among our study population and by being in a university hospital where obstetrics specialists are trained.

The high episiotomy rate seen among those women who received oxytocin can be explained by the greater incidence of instrumental births among them and it is similar to those findings by another study that analyzed adverse birth outcomes in low-risk nulliparous women under oxytocin administration [14].

In terms of PPH, there were no statistically significant differences between women who received oxytocin in labor and those who did not. After adjusting for the indication to oxytocin administration, results showed a positive association between IOL and PPH; according to Rousseau et al., this association of risk of PPH and oxytocin administration may be related to the dose administered [18], which leads us to think our results are because high doses of oxytocin are always more associated with IOLs.

With regard to neonatal complications, our results show that oxytocin administration is not a risk factor for admission to NICU. However, when we conducted a subanalysis for the reason of administration we found that when oxytocin was administered to induce labor, there was no statistically significant increased risk association of admission to NICU. These findings coincide with others [19,44].

The risk of obtaining a pH ≤ 7.20 in the umbilical artery is higher when oxytocin is administered during labor. These results are similar to those found by Hidalgo-Lopezosa et al. [45]. Jonsson et al., in 2008, associated neonatal acidemia with uterine hyperstimulation and oxytocin administration [21] and Bakker et al., in 2007, also found an association between excessive uterine activity and low pH values [22]; our study did not collect data on uterine activity, therefore this association was been demonstrated.

With regard to the other neonatal outcomes we measured in our study (Apgar test at 1 and 5 min of life, the need for neonatal resuscitation and admission to NICU), we did not find any association with the use of oxytocin. However, other studies using different methodologies have found a relationship between oxytocin administration during labor and lower Apgar results, greater need for neonatal resuscitation and admission to NICU [7,15,23,46].

Several studies have demonstrated that identifying the optimal time for commencing oxytocin, captured in guidelines that also stipulate the appropriate dose to employ, can reduce the risk of neonatal complications such as low Apgar scores and admission to NICU, as well as maternal complications, including cesarean section [17,46,47].

### 4.1. Strengths and limitations

Our study offers some tools to understand that oxytocin administration during low-risk labor may not be an innocuous intervention and can result in a series of complications for both the mother and neonate.

Following the analysis of our results, we identify the need to continue researching the use of different unnecessary labor interventions which can be considered endemic in our practice area. This would allow us to develop strategies to provide maternity care that is more aligned with the promotion of labor physiology and more in synchronicity with women’s wishes [48,49].

The main limitation of our study lies in its observational methodology, given the need to rely on previously recorded variables. This limits our ability to establish causal relationships, such as the use of EA and its association with oxytocin administration. We were not able to establish whether EA was used more in the presence of oxytocin because the latter makes labor more painful, or whether EA led to a higher need in oxytocin use because it interfered with effective uterine contractions.

There is a paucity of data examining the complications arising as a result of administering oxytocin to low-risk women, and it was, therefore, difficult to compare our results with other studies.

The reason why oxytocin is employed both for IOL and AOL is not routinely recorded in obstetric records in Spain. Therefore, we were unable to ascertain whether the use of oxytocin was justified or not.

### 4.2. Implications for the Practice and Research

Given these results, it would seem reasonable to suggest that optimizing safety and standardization of oxytocin administration criteria is required as a matter of priority [50,51]. In particular, evidence-based guidelines on the appropriate oxytocin use and its dosage would contribute to improve outcomes and reduce unwanted ones. Outcome indicators that allow for ongoing assessment of adherence, effectiveness and review based on the latest research would also be beneficial [52].

However, our results also show excessive and possibly inappropriate use of other labor interventions in low-risk women, with rates of episiotomies, EA and ARM greater than those recommended in the national and international literature [11,12,13]. These interventions during normal childbirth have been steadily increasing over the last few decades in the developed world in the absence of evidence of their benefit. The contrary is happening: the scientific evidence demonstrates that this over intervention is causing harm to mothers, babies and families [25].

To optimize maternity care provision for low-risk women, the medicalization and hierarchical organization of childbirth care provision must be tackled as these factors obstruct the effective use of midwives’ skills in providing care for them, hinders care provision based on the holistic needs of women and prevents them from voicing their opinions and preferences [53].

However, an improvement in the use of oxytocin administration and other obstetric interventions should not happen as an isolated practice change but needs to be embedded within a culture of approaching labor care from a more physiological perspective, where respect for the process and a belief in the ability of women’s bodies to successfully give birth with minimal intervention is intrinsic to the philosophy of care of every maternity unit. Furthermore, a discourse around respectful maternity care which is gathering momentum around the world [54] needs to be had across Spanish birth settings.

## 5. Conclusions

The results of this study show that oxytocin administration for low-risk women may not be an innocuous intervention. Its use must be adequately justified through consensual guidelines based on the latest national and international research and recommendations.

It appears that the use of oxytocin for low-risk women surpasses the levels recommended by the National Strategy [5]. The extent of complications arising as a result of the use of oxytocin appears to be lower than in other studies, possibly because of the lower dosages employed in our setting. Nonetheless, efforts need to be focused on ensuring more judicious use of the drug.

Oxytocin is thought to be used to reduce detrimental labor outcomes, but our results, which confirm other studies’ findings, show an association between oxytocin administration and a higher number of instrumental births, cesarean sections, use of episiotomy and EA. Neonatal outcomes are also adversely affected by the use of oxytocin, with a greater percentage of umbilical artery pH ≤ 7.20 when it is given to induce labor.

## Figures and Tables

**Figure 1 ijerph-18-04375-f001:**
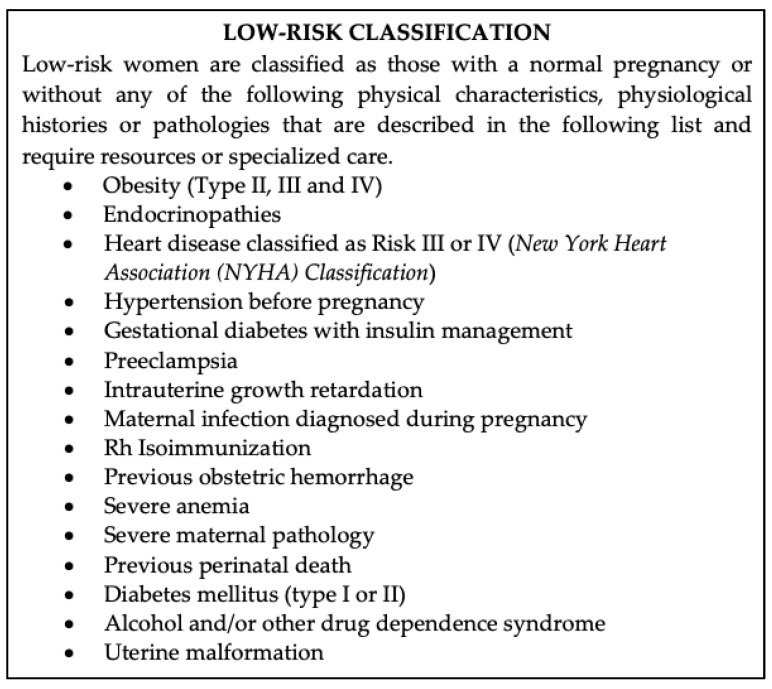
Low-risk classification.

**Figure 2 ijerph-18-04375-f002:**
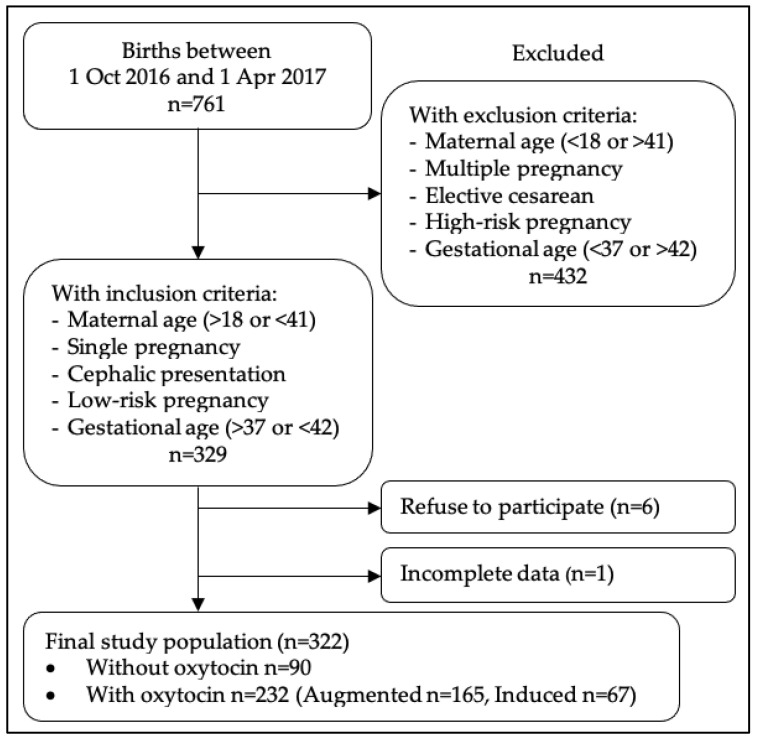
Flow chart of patient selection.

**Table 1 ijerph-18-04375-t001:** Maternal and neonatal characteristics.

	with Oxytocin (*n* = 232)		without Oxytocin (*n* = 90)			
	*n*	%	*n*	%	Total	*p-*Value
Maternal age in years (mean/sd)	31/5.14		30/5.4		322	0.27
Length of the first stage of labor * (median/IQR)	345/235		240/236		301	0.01
Length of the second stage of labor* (median/IQR)	40/66		24/43		301	<0.01
Nationality						
Spanish	152	65.8%	44	48.9%	197	
European	13	5.6%	3	3.3%	16	
South-Central American	17	7.4%	9	10.0%	26	
African	45	19.5%	31	34.4%	76	
Asian	4	1.7%	3	3.3%	7	0.03
Educational level						
Primary studies	74	31.9%	39	43.3%	113	
Secondary studies	84	36.2%	21	23.3%	105	
University studies	66	28.4%	25	27.8%	91	
Unknown	8	3.4%	5	5.6%	13	0.09
Gestational age						
37 weeks	12	5.2%	13	14.4%	25	
38 weeks	30	12.9%	11	12.2%	41	
39 weeks	59	25.4%	28	31.1%	87	
40 weeks	71	30.6%	27	30.0%	98	
41 weeks	60	25.9%	11	12.2%	71	0.01
Parity						
Nulliparous women	140	60.3%	31	34.4%	171	
Multiparous women	92	39.7%	59	65.6%	151	<0.01
Birthweight **						
<3000	43	18.5%	17	18.9%	60	
3000–3500	106	45.7%	50	55.6%	156	
3501–4000	68	29.3%	20	22.2%	88	
>4000	15	6.5%	3	3.3%	18	0.30

* Time expressed in minutes. ** Weight expressed in grams. Sd: Standard deviation; IQR: Interquartile range.

**Table 2 ijerph-18-04375-t002:** Crude and adjusted OR for birth outcomes with oxytocin as a predictor.

	Crude OR				Adjusted OR			
	OR	95% CI		*p-*Value	Adj OR	95% CI		*p*-Value
		Lower	Upper			Lower	Upper	
Cesarean section	3.93	0.90	17.21	0.07	4.81 *	1.80	12.82	<0.01
Instrumental birth	3.64	1.58	8.39	<0.01	3.34 **	1.45	7.67	<0.01
Episiotomy	6.13	3.26	11.53	<0.01	3.79 ^†^	2.20	6.52	<0.01
Postpartum hemorrhage	5.28	0.68	4.99	0.11	1.55 ^‡^	0.49	4.91	0.46
Length of first stage labor (hh:mm)	0:46	0:00	1:32	0.05	0:46 ^§^	−0:02	1:35	0.06
Length of second stage labor (hh:mm)	0:18	0:04	0:31	0.01	0:18 ^§^	0:04	0:31	0.01

* Analyses were adjusted for parity, epidural analgesia, birth weight and gestational age. ** Analyses were adjusted for parity, epidural analgesia and birth weight. ^†^ Analyses were adjusted for parity, epidural analgesia and vaginal birth. ^‡^ Analyses were adjusted for parity, epidural analgesia and gestational age. ^§^ Analyses were adjusted for parity and epidural analgesia. CI: Confidence Interval; OR: Odds Ratio; aOR: adjusted Odds Ratio.

**Table 3 ijerph-18-04375-t003:** Adjusted OR for birth outcomes with oxytocin as a predictor by parity and oxytocin indication.

	Adjusted OR by Parity								Adjusted OR by Oxytocin Indication							
	Nulliparous				Multiparous				IOL				AOL			
	aOR	95% CI		*p*-Value	aOR	95% CI		*p*-Value	aOR	95% CI		*p*-Value	aOR	95% CI		*p-*Value
		Lower	Upper			Lower	Upper			Lower	Upper			Lower	Upper	
Cesarean section	1.08	0.88	1.32	0.54	-	-	-	-	2.20	1.64	2.96	<0.01	1.21	0.85	1.74	0.40
Instrumental birth	1.14	1.00	1.30	0.09	1.30	0.89	1.89	0.29	1.69	1.17	2.44	0.02	1.39	1.18	1.65	<0.01
Epidural analgesia	4.27	1.24	14.83	<0.01	9.63	3.24	28.61	<0.01	4.54	1.36	15.11	0.01	13.84	3.55	53.92	<0.01
Episiotomy	1.75	1.06	2.86	<0.01	1.66	1.18	2.33	<0.01	2.82	1.55	5.14	<0.01	2.12	1.53	2.95	<0.01
Postpartum hemorrhage	-	-	-	-	1.39	0.95	2.03	0.25	2.01	1.34	3.01	0.04	1.40	1.09	1.79	0.12
Length of first stage labor (hh:mm)	−0:08	−1:20	1:03	0.81	0:37	−0:23	1:38	0.23	0:18	−0:45	1:23	0.57	0:56	0:06	1:46	0.03
Length of second stage labor (hh:mm)	0:14	−0:10	0:39	0.26	0:01	−0:07	0:10	0.72	0:07	−0:08	0:24	0.33	0:23	0:10	0:36	<0.01

IOL: Induction of labor; AOL: Augmentation of labor; CI: Confidence Interval; aOR: Adjusted Odds Ratio.

**Table 4 ijerph-18-04375-t004:** Crude and adjusted OR for neonatal outcomes with oxytocin as a predictor by type of vaginal birth and oxytocin indication.

	Adjusted OR by Type of Vaginal Birth								Adjusted OR by Oxytocin Indication							
	Instrumental Birth				Spontaneous Birth				IOL				AOL			
	aOR	95% CI		*p*-Value	aOR	95% CI		*p*-Value	aOR	95% CI		*p-*Value	aOR	95% CI		*p*-Value
		Lower	Upper			Lower	Upper			Lower	Upper			Lower	Upper	
1 min Apgar test < 7	0.05	0.01	0.43	0.01	1.52	0.30	7.70	0.61	0.55	0.14	2.20	0.40	0.45	0.15	1.38	0.16
Neonatal resuscitation	0.51	0.05	5.36	0.58	0.75	0.12	4.55	0.75	1.34	0.26	6.85	0.73	0.91	0.21	3.91	0.90
Admission to NICU	0.65	0.07	6.57	0.72	0.24	0.06	0.97	0.05	1.18	0.41	3.42	0.77	0.32	0.10	1.02	0.05
Umbilical artery Ph ≤ 7.20	2.14	0.22	20.94	0.51	2.72	0.98	7.55	0.05	3.71	1.32	10.46	0.01	3.12	1.23	7.95	0.02

IOL: Induction of labor; AOL: Augmentation of labor; CI: Confidence Interval; aOR: Adjusted Odds Ratio.

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
