# Peer review of "Oxytocin Administration in Low-Risk Women, a Retrospective Analysis of Birth and Neonatal Outcomes"

_ijerph, 2021, doi:10.3390/ijerph18084375_

Round 1

Reviewer 1 Report

This is a retrospective analysis about a topic of major interest. The statistical calculations are thoroughly done and the results are presented well. However, a major pitfall lies in the nature of the study being a retrospective analysis: It is nearly impossible to discern whether the use of oxytocin led to a worse outcome of birth or if there was already a prexisting risk for an adverse outcome in these women which led to the use of oxytocin. In line 487 you say it's a paradox that the administration of oxytocin leads to a poorer birth outcome. I don't think your study allows you to draw the conclusion that the use of oxytocin is causative for this. The study shows that oxytocin is used more often in these cases but the primary reason for the poor outcome is unclear. As a consequence I would recommend to use more restrained wording at several sections: E.g. in the abstract, conclusions: please say that the results of this retrospective analysis show that the administration of oxytocin may results in worse birth outcomes. In line 432: How can you say that these complications can be avoided when not using oxytocin? Your retrospective analysis does not support this.

Firstly, I think it is absolutely necessary to exclude the women who recieved oxytocin for IOL. These women cannot be compared with women with spontaneous start of labour. Furthermore it needs to be clarified in more detail how IOL was done: did you only use oxytocin or did you also use prostaglandins? Did you routinely perform ARM to induce labour?

In Fig. 1 there seems to be a mistake: As I understand these factors define women with a high-risk pregnancy? In Fig. 2 it would be nice to show how many women of the final study pop. recieved oxytocin and how many IOL.

There is a surprisingly high rate of instrumental birth and - even more surprising - these cases seem to be associated with a lower rate of one-minute-Apgar <7 which is very contradictory to my personal experience. Please comment on that.

Reviewer 2 Report

It need s to clearer that these were exclusions

with a normal pregnancy or without any of 
the following physical characteristics

LOW-RISK CLASSIFICATION
Low-risk women are classified as those with a normal pregnancy or with  any of 
the following physical characteristics, physiological history or pathology that are 
described in the following list and does not require resources or specialized care.
• Pelvic abnormalities
• Low height (<1.45 meters)
• BMI:
- Overweight grade I (BMI: 25.0-26.9)
- Overweight grade II (BMI: 27.0-29.9)
- Type I obesity (BMI: 30.0-34.9)
• Unwanted pregnancy
• Women without prior vaccination of:
- Tetanus vaccine
- Pertussis vaccine
- Flu shot
- Varicella vaccine
• Great multiparous (> 4 previous births)
• Short intergenic period (<12 months)
• History of intrauterine growth retardatio
